# Towards an Arabic Sign Language (ArSL) *corpus* for deaf drivers

Samah Abbas[1], Hassanin Al-Barhamtoshy[2] and Fahad Alotaibi[3]

[1] Management Information Systems Department, Faculty of Economics and Administration, King Abdul Aziz University, Jeddah, Mecca, Saudi Arabia
[2] Information Technology Department, Faculty of Computing & Information Technology, King Abdul Aziz University, Jeddah, Mecca, Saudi Arabia
[3] Information System Department, Faculty of Computing & Information Technology, King Abdul Aziz University, Jeddah, Mecca, Saudi Arabia



Corresponding author
Samah Abbas, sabbas@kau.edu.sa

## ABSTRACT

Sign language is a common language that deaf people around the world use to communicate with others. However, normal people are generally not familiar with sign language (SL) and they do not need to learn their language to communicate with them in everyday life. Several technologies offer possibilities for overcoming these barriers to assisting deaf people and facilitating their active lives, including natural language processing (NLP), text understanding, machine translation, and sign language simulation. In this paper, we mainly focus on the problem faced by the deaf community in Saudi Arabia as an important member of the society that needs assistance in communicating with others, especially in the field of work as a driver. Therefore, this community needs a system that facilitates the mechanism of communication with the users using NLP that allows translating Arabic Sign Language (ArSL) into voice and *vice versa*. Thus, this paper aims to purplish our created dataset dictionary and ArSL *corpus* videos that were done in our previous work. Furthermore, we illustrate our *corpus*, data determination (deaf driver terminologies), dataset creation and processing in order to implement the proposed future system. Therefore, the evaluation of the dataset will be presented and simulated using two methods. First, using the evaluation of four expert signers, where the result was 10.23% WER. The second method, using Cohen's Kappa in order to evaluate the *corpus* of ArSL videos that was made by three signers from different regions of Saudi Arabia. We found that the agreement between signer 2 and signer 3 is 61%, which is a good agreement. In our future direction, we will use the ArSL video *corpus* of signer 2 and signer 3 to implement ML techniques for our deaf driver system.

## INTRODUCTION

Deaf people use sign language to communicate with their peers and normal people who know their sign language. Therefore, sign language is the only way to communicate with deaf people, although it receives even less attention from normal people. Furthermore, sign language is not a unified language among deaf people around the world,

as each country has its own sign language. For example, American Sign Language (ASL), British Sign Language (BSL), Australian Sign Language (Auslan), Indian Sign Language (ISL) and Arabic Sign Language (ArSL). In particular, the Arab countries that use ArSL, such as the Gulf, Al-Sham, and some North African Arab countries, have similarities and differences in sign language. The reason for this is the differences between in dialects. As a result, deaf people face problems communicating with the community in many aspects while they are working or practicing their daily lives, such as health, education, and transportation. One of the solutions to these problems is the use of a sign language interpreter, a person who knows sign language and can interpret it to normal people. However, this solution is not optimal because of loss of privacy and professional independence (*Forestal, 2001*; *Broecker, 1986*). Researchers have therefore developed certain technologies that allow for the provision of a computer interpreter instead of a human. For example, sign language recognition technology, machine translation (MT). In addition, some Arab researchers and organizations, such as the Arab League Educational, Cultural and Scientific Organization (ALECSO), have made efforts to unify ArSL by introducing the first dictionary in 1999 (*Al-Binali & Samareen, 2009*).

In Saudi Arabia, deaf people have some difficulty communicating with others. In particular, when deaf people are driving a vehicle and non-deaf people are sitting as passengers. The non-deaf passenger cannot understand the deaf driver. Also, a non-deaf passenger cannot describe the location needed by the deaf driver. However, there are many mobile applications that can be used to facilitate communication, such as Tawasol and Turjuman, but they still do not provide real-time translation (*Team Mind Rockets, 2017*; *Al-Nafjan, Al-Arifi & Al-Wabil, 2015*). In addition, non-deaf passengers or drivers do not necessarily download the sign language translation application and use it only once. In the area of deaf drivers, Saudi Arabia faces a lack of technology that can improve communication between deaf drivers and non-deaf passengers. To our knowledge, no research has yet been conducted to introduce a solution in the area of deaf drivers in Saudi Arabia. In addition, there is a lack of *corpus* of ArSL videos being constructed in the transportation domain, especially for deaf drivers.

The purpose of this research is to publish our dataset as a continuation of previous work (*Abbas, 2020*). Also, we aim to publish our *corpus* in order to implement ML as future work. This paper is organized as follows: The second section illustrates some research work done to create videos and images of the standard *corpus* of sign language in each country as a database. In order to conduct their experiments to implement some systems that help deaf people in specific areas. We have also mentioned some research done in the Arab countries. In order to create an ArSL *corpus* by focusing on certain areas. The third section illustrates the linguistic background of ArSL, speech recognition and machine translation (MT) systems, while the fourth section focuses on the design architecture of ArSL. In the fifth section, we discuss the processing and data collection modules and evaluation videos of the deaf driver *corpus* that necessitated the implementation of ArSL in the deaf driving context. Finally, we illustrate future research directions.

## RESEARCH QUESTION

Our research aims to answer these two questions:

1. Can we create a data *corpus* and video *corpus* for deaf drivers?
2. Can we evaluate the video *corpus* created of deaf drivers?

## LITERATURE REVIEW OF ARSL RECOGNITION AND MT SYSTEM

This literature review illustrates various standard sign language corpora for some countries of the world. In fact, when we start designing and developing a system for deaf people, we need the essential corpora of videos or images. In addition, these corpora are considered as a reference database that must be rich with a large and complete volume of standard sign language. They will help researchers to implement machine learning techniques and develop specific systems for deaf people.

Starting with ASL, researchers developed approximately 2,576 videos that included movements, hand shapes, and sentences (*Martinez et al., 2002*). Another researcher collected and annotated videos around a *corpus* of 843 sentences from Boston University and RWTH Aachen University (*Dreuw et al., 2008*). In Britain, approximately 249 participants completed a conversational dataset of the BSL video *corpus* (*Schembri et al., 2013*). In Indian sign language, researchers implement 1,440 gestural videos of IPSL by nine Indian signers (*Kishore et al., 2011*). In Brazil, they have created the LIBRAS-HCRGBDS database of the Libras video *corpus* which has approximately 610 videos by five signers (*Porfirio et al., 2013*). In Korean Sign Language (KSL), the researchers developed 6,000 vocabulary words in the KSL *corpus* database (*Kim, Jang & Bien, 1996*). In Arabic Sign Language (ArSL), 80 Arabic signers implemented a sensor-based dataset for 40 sentences (*Assaleh, Shanableh & Zourob, 2012*). In addition, some researchers have produced a *corpus* of 500 videos and images. It contains hand shapes, facial expressions, alphabet, numbers. In addition, the movement in simple signs, continuous sentences, and sentences with lip movement were performed by 10 signers. These corpora are called SignsWorld Atlas. However, the performed *corpus* still does not cover all words and phrases as a database (*Shohieb, Elminir & Riad, 2015*). Recently, the researchers led their efforts to build the KArSL database as a comprehensive ArSL reference database for numbers, letters, and words. They used Kinect devices to make these ArSL videos available to all researchers (*Sidig et al., 2021*).

Specifically, in ArSL, researchers are dedicating their efforts to improving the focus of accuracy in a specific domain, such as medicine and education. Some researchers used 150 video signs from the Java programming domain in their experiment (*Al-Barhamtoshy, Abuzinadah & Allinjawi, 2019*). In the same educational field, two researchers introduced an intelligent system for deaf students based on the image. This system helped the deaf student in the educational environment. Therefore, they created the dictionary that needs it in life activities and academic environments (*Mohammdi & Elbourhamy, 2021*). Some of them performed some sentences in Arabic language without

focusing on a specific field. Some focused on the jurisprudence of prayer and their symbols as datasets (*Almasoud & Al-Khalifa, 2012*). Other researchers used 600 phrases in the health domain (*Luqman & Mahmoud, 2019*). Similarly, some researchers have implemented their research using the data created in the field of education (*Alfi, Basuony & Atawy, 2014*; *Almohimeed, Wald & Damper, 2011*). In our research, we focus on a different domain that can help deaf people in the work environment, such as driving a cab. The *corpus* we created will help deaf drivers to communicate with non-deaf passengers while deaf people drive their cars.

## ARSL AND LINGUISTIC BACKGROUND

ArSL shows huge complexities in phonology, morphology, and structure, which is not the case for other sign languages. These complexities are explained below.

### Phonology of ArSL

Phonemes are mental representations, which are just a way to empty what is inside the brain. Phonemes are made up of four elements: (1) The shape of the hand. (2) Orientation of the hand. (3) Position of the hand. (4) Direction of the hand in motion (*Schechter, 2014*). In phonology, these four elements are known as hand features (MFs). In particular, in Sign Language, there are MFs and also Non-Manual Features (NMFs) are involved. NMFs refer to emotional parts of the body, for example, lip motion, facial expression, shoulder movement, head movement, eyelids, and eyebrows. In general, in ArSL, we use both MFs and NMFs to give the correct meaning, called essential NMFs. On the other hand, if the signer uses just only the MFs, the meaning will turn into another meaning that the signer did not mean to express (*Johnston & Schembri, 2007*).

### Morphology and structure of ArSL

The rules of grammar in ArSL are not the same as those in Arabic. The differences are: verb tenses, differences between singular and plural, rules for prepositional and adverb and gender signs. Regarding sentence structure, ArSL uses only the Subject-Verb-Object (SVO) structure instead of the SVO, OVS and VOS structures (*Luqman & Mahmoud, 2019*).

## ARSL AND DESIGN ARCHITECTURE

New technologies that support communication have a significant impact on human life. For deaf people, the developers and researchers have tried to use some new technologies to make the life of deaf people easier by developing automated systems that can help them to communicate in various aspects of their life with others. In this section, we will illustrate a brief explanation of some of the techniques used in order to implement the automated system for better communication between the community and among themselves.

### Machine translation

Machine Translation (MT) is a standardized name for the system that relies on computer analysis to translate between two languages. It is used for text and speech using artificial intelligence (AI) and natural language processing (NLP). For example, translating from

source "English" to target "Arabic" (*Pradhan, Mishra & Nayak, 2020*; *Verma, Srivastava & Kumar, 2015*). MT is also used to translate text or speech into video sign language or avatar. There are several approaches to MT that can be used depending on what we need to translate and what constitutes a better translation. One of these approaches is a direct translation, without regard to grammar rules. To improve the quality of MT, the rule-based approach was introduced, which consists of parsing the source and target language. Another approach is *corpus*-based, which deals with massive data containing sentences. There are also knowledge-based approaches, which take into account the understanding of the source and target text in the context of linguistic and semantic knowledge. Finally, there is google translation which is developed by Google (*Verma, Srivastava & Kumar, 2015*).

## Arabic speech recognition

Speech Recognition (SR) is a computerized system that converts speech into text or signs. This system is used to communicate between humans and machines. It is also known as automatic speech recognition. Machine-based SR is a complicated task due to differences in dialects, contexts, and speech styles. To reduce this complexity in SR, the system can exploit the repetition and structure of the token speech signal as multiple sources of knowledge. SR sources are constructed based on knowledge of phonology, syntax, phonemes, grammar, and phrases (*Katyal, Kaur & Gill, 2014*; *Chow et al., 1987*). In addition, the SR system has several classifiers. (1) The speech utterance, which is composed of separate, connected, continuous and spontaneous speech. (2) The speaker model, which contains one of two dependents that was designed for a specific speaker or independently for different speakers. (3) The size of vocabulary (*Vimala & Radha, 2012*).

In terms of the SR process, there are four steps in which SR can be implemented. Analyze the speech signal, then extract the feature using different techniques to identify the vector, such as MFCC (Mel-frequency cepstral coefficient). Then, we build a model using different techniques like HMMs with the training dataset. In the last step, we test the model in the matching setting, taking the dissection and measuring the performance based on the error rate (*Saksamudre & Deshmukh, 2015*; *Yankayiş, Ensari & Aydin, 2018*) as shown in Fig. 1.

## ArSL gestures recognitions

Gesture recognition is defined as the ability of the computer to understand gestures and execute commands based on the gestures made. The first gesture recognition system was introduced in 1993 as a kind of user interface for perceptual computing that helps capture human gestures and transfer them into commands using a computerized system. It is used with many technologies, especially in the field of games, such as X-box, PlayStation, and Wii Fit. These games use Just Dance and Kinect Sports, which recognizes the hand and certain body parts (*Schechter, 2014*; *Darrell & Pentland, 1993*).

In the sign language domain, the gesture recognition system uses the following processes: (1) Recognize the deaf signs. (2) Analyze the sign. (3) Converting that sign into meaningful text (words or phrases), voice, or expressions that non-deaf can understand.

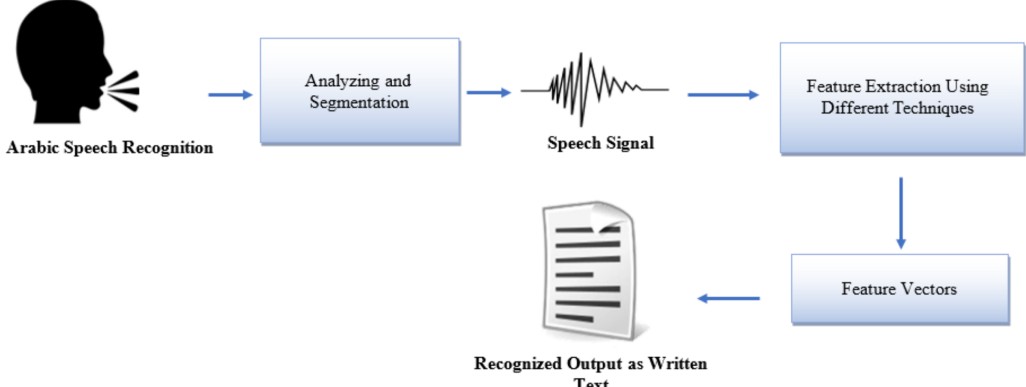

**Figure 1  A diagram for recognizing spoken the Arabic language to Arabic written text (*Abbas, 2020*).**

In addition, there are two main methods for gesture recognition in ArSL, namely vision-based devices (video or image) and wearable-based devices. Each of them has advantages and disadvantages. One advantage of wearable devices is that there is no need to search for background and lighting. The disadvantage is that they interfere with movement. On the other hand, the advantage of vision-based technology is the ease of movement, while the disadvantage is the effect of changing the background and lighting (*Ambar et al., 2018*; *Paudyal et al., 2017*).

In addition, each has different processes and techniques. In the vision-based process, one or more cameras are the main tools that must be available to use this method. On the other hand, the wearable-based method depends on certain types of equipment and computers mainly. In terms of process, the wearable-based method spells the alphabet by reading the specific information in each sensor of the finger or glove joint. However, the vision-based method has certain steps, which are the follows:

- Image capture, which consists in using a camera to collect data (building the *corpus*) and analyze the collected images.
- Pre-processing, which consists of preparing the images and identifying the information according to color (segmentation).
- Feature extraction uses some techniques like root mean square (RMS) to identify the feature vector.

Classification that classifies based on the feature vector to build the model (*Mahmood & Abdulazeez, 2019*; *Derpanis, 2004*). The vision-based gesture recognition process is shown in the Fig. 2.

## DEAF DRIVER *CORPUS*

For creating a deaf driver *corpus*, we divided the processes that we will use for this creation into four modules. The four modules are preprocessing, recording, assessment, and validation modules. These high-level and low-level approaches are represented in (Figs. 3 and 4).

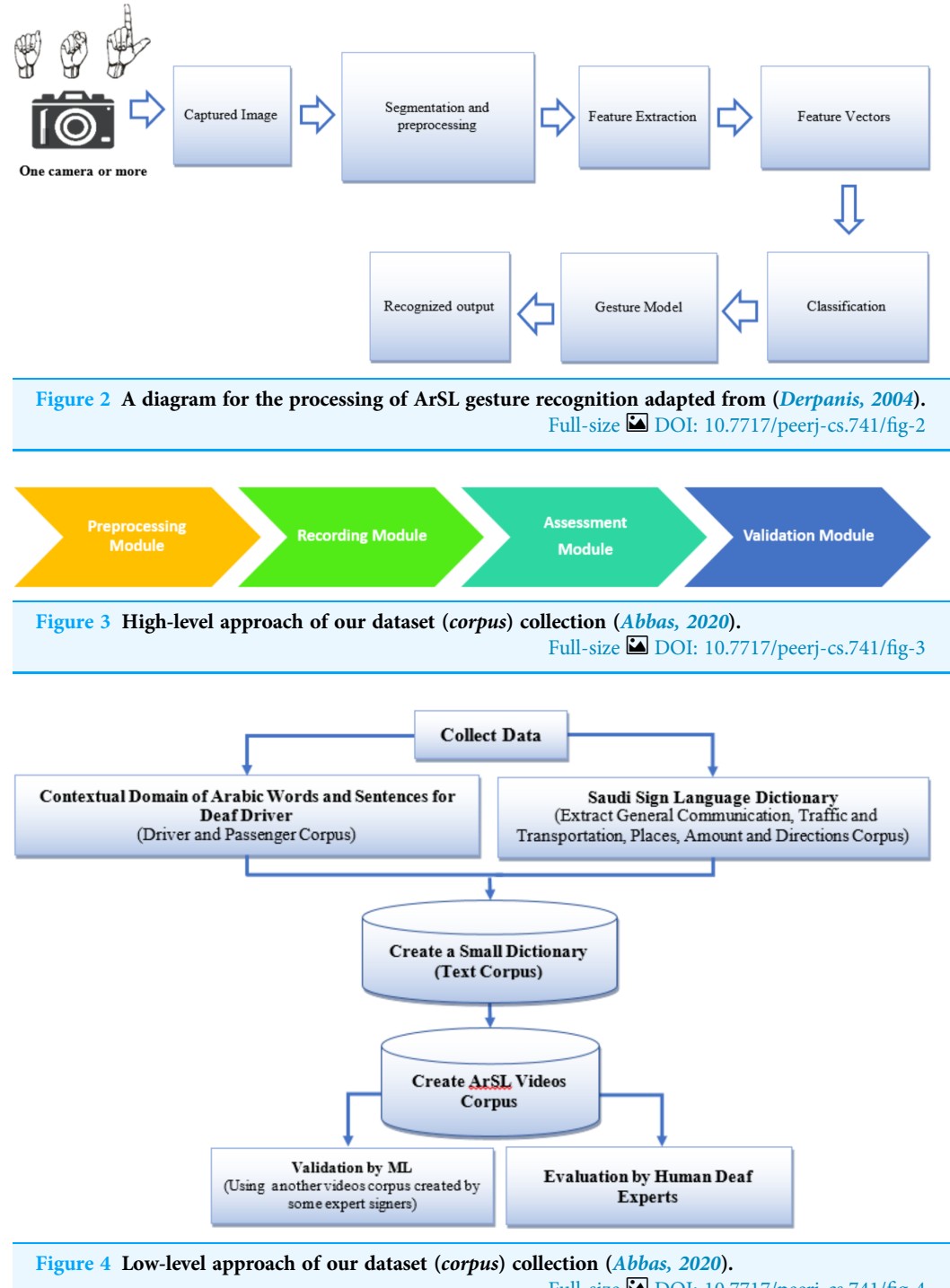

**Figure 2** A diagram for the processing of ArSL gesture recognition adapted from (*Derpanis, 2004*).

**Figure 3** High-level approach of our dataset (*corpus*) collection (*Abbas, 2020*).

**Figure 4** Low-level approach of our dataset (*corpus*) collection (*Abbas, 2020*).

## Data collection, creation and annotation

In this section, we explain our data collection, data determination (deaf driver terminologies) and data set creation and processing to create our video *corpus* through the preprocessing and recording module.

**Table 1** An example of our created dictionary (*Abbas, 2020*).

| Word/phrase/sentences |
|---|
| **Section 1: Welcome** |
| 1. Salam Aleikum (peace be upon you) |
| 2. How are you? |
| **Section 2: Directions** |
| 1. Left |
| 2. Right |
| **Section 3: Place** |
| 1. School |
| 2. Deaf association |
| **Section 4: Traffic and transport** |
| 1. Driving license |
| **Section 5: Driver** |
| 1. We have arrived |
| **Section 6: Passenger** |
| 1. I do not have cash to pay the amount |

In the preprocessing module, we first collected data at two levels: a word or phrase level dataset and a sentence level dataset to create a small Arabic dictionary. This dictionary is divided into eight sections (categories). (1) Welcoming "Salam Alaikum and How are you?". (2) Directions "Left and Right". (3) Place "School and Deaf Association". (4) Traffic and Transportation "Driver's license and Traffic lights". (5) Sentences used by deaf drivers when they need to talk with their passengers, *e.g.*, "We have arrived". (6) Sentences used by passengers when they need to talk with their deaf drivers, *e.g.*, "I do not have cash to pay the amount". (7) General Words "No and Yes and In and On". (8) Amount "Dollar, Riyal and 2 Riyals until 100 Riyals" (*Abbas, 2020*).

The total number of words and phrases (sentences) is 215. Some of these words and sentences fall under the general communication, collected from the Saudi Sign Language Dictionary, 2018 edition (*Saudi Association for Hearing Impairment, 2018*). Some of them are collected in the contextual domain of the normal conversation that was done between Saudi cab drivers and passengers. We mean in the contextual domain is each country has its own contextual domain in the payment process. For example, Saudi Arabia has its own curranty which is Riyal. Table 1 shows a part of the Arabic dictionary we created.

In the recording module, we made our videos at a rate of about 30 FPS (frames per second), by a camera for our ArSL *corpus*. These videos were taken by one of the ArSL expert signers who is not deaf. Figure 5 shows an ArSL *corpus* captured by a non-deaf expert signer. In addition to that, we made video captures with three expert signers from different occasions in Saudi Arabia. Two of them are totally deaf, while the third is hard of hearing. Figure 6 represents an ArSL *corpus* captured for each deaf expert. To record 215 words containing simple phrases and signs, we took about 45 min continuously with the non-deaf expert signer. However, the three deaf expert signers took approximately 50 min. The expert signers used only one hand if it was appropriate for the context of the

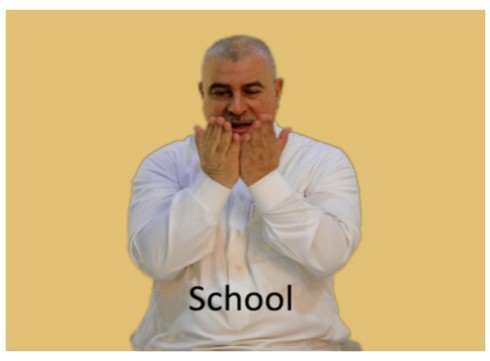

**Figure 5** One captured video from our ArSL video *corpus* (*Abbas, 2020*).

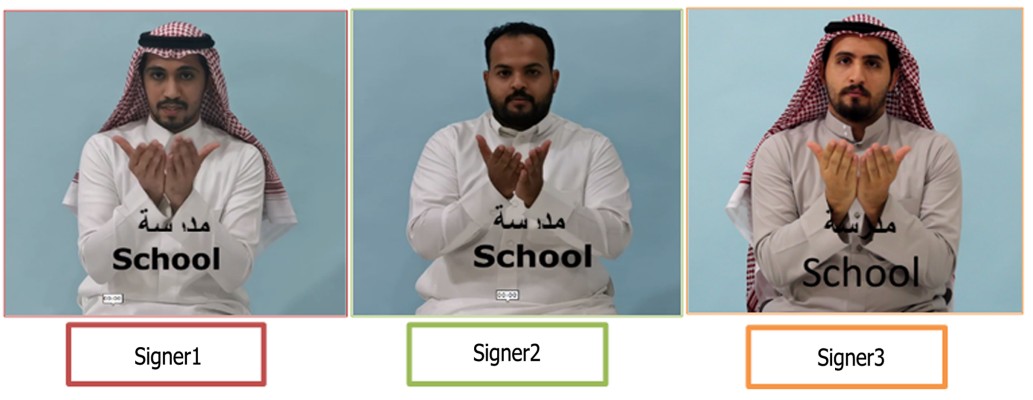

**Figure 6** Three captured ArSL video *corpus*.     

deaf driver, unless the sign required the use of both hands. Next, we segmented the video using VEGAS (*VEGAS, 2022*) video editing software (we segmented each sign containing words or phrases that matched our dictionary into a single video, with a total number of 215 videos). To support future work, we added Arabic audio to each video and labeled it with Arabic text that refers to the same recorded ArSL.

## Dataset evaluation

These collected and created data will be used for the evaluation and validation of our ArSL *corpus*. In this section, we have explained the evaluation module while the validation will be done in the future work.

In the evaluation module, we evaluated the video of each *corpus* generated based on our created dictionary using two techniques. First, we used a human expert evaluation technique. Second, we used a statistical method of measuring the agreement between three deaf experts signing in ArSL that we recorded.

In the first technique, which is a human expert evaluation, we made the evaluation based on the views of four participants who are experts in ArSL. One of them is completely deaf and the second is hard of hearing. Two other participants are not deaf, they work in a deaf club and are experts. We used the quantitative approach (questionnaire) and divided

**Table 2 An example of evaluation with one expert (*Abbas, 2020*).**

| Section one: Welcome | | | | |
|---|---|---|---|---|
| Correction if the translation is wrong: The video of sign needs. | | | Video evaluation based on the related word (phrase) or sentences. × or √ | Word\phrase\sentences |
| Deleting | Replacing | Adding | | |
| | | | √ | 1-Salam aleikum |
| | | | √ | 2-Waleikum salam |
| | √ | | × | 3-Good morning |

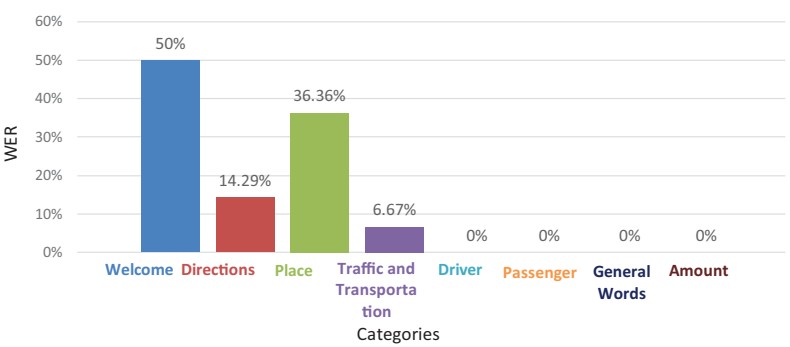

**Figure 7 The deaf experts' evaluations results for each category using (WER) (*Abbas, 2020*).**

it into two sections. The first section was a demographic questionnaire (gender-age-education level-whether he/she is deaf or not). In the second section, we introduced an evaluation of the video based on the word (phrase) or related sentence. We attached each video to each related phrase or word. Participants were asked to evaluate the 215 sign videos to see if the video recorded for each word or phrase was correct or not. If not correct, it meant that the video was not related to that particular word or phrase. We asked the four participants to choose one of the types of correction they had to make for each video (add-replace-delete). The evaluation method is explained in (Table 2) using some videos evaluated by one of the experts.

The evaluation results of the deaf experts based on the word error rate (WER) for each category (section) such as welcome, directions, and place are explained in Fig. 7. This means that for each category, the evaluation result of these videos was wrong. Also, they should be replaced with correct videos.

As can be seen in Fig. 7, first, the hospitality category has a high percentage of WERs, at 50%. Second, the location category has a percentage of about 36%. Thus, we need to reduce the WER in our video *corpus* that is related to these two categories (sections) in order to improve the communication between drivers and deaf passengers and also to describe the correct location for the passenger's destination.

The total WER of our video *corpus* was 10.23% as shown in Table 3.

**Table 3 Total word correct and error rate for our created video *Corpus* (*Abbas, 2020*).**

|  | Correct (%) | Wrong (%) |
|---|---|---|
| Signs' videos *corpus* | 89.8 | 10.2 |

**Table 4 Categories of Cohen's Kappa interpretation (*Viera & Garrett, 2005*).**

| Type of agreement | Kappa (K) |
|---|---|
| No agreement | $K \leq 0$ |
| Poor agreement | $0.01 \leq K \leq 0.20$ |
| Fair agreement | $0.21 \leq K \leq 0.40$ |
| Moderate agreement | $0.41 \leq K \leq 0.60$ |
| Good agreement | $0.61 \leq K \leq 0.80$ |
| Perfect agreement | $0.81 \leq K \leq 1.00$ |

To resolve the *corpus* of erroneous ArSL videos, we again captured these videos based on the corrected signs that the raters explained to us.

In the second technique of measuring the agreement between the signs of each of the two deaf experts, we used Cohen's Kappa criterion using Eq. (1).

$$K = \frac{P_0 - P_e}{1 - P_e} \tag{1}$$

whereas $P_0$ represent the number of observed proportional agreement between two variables by using Eq. (2).

$$P0 = \frac{1}{n} \sum_{i=1}^{g} fii \tag{2}$$

Number of agreements that expected by chance represented by *Pe* and the formula represents as Eq. (3).

$$P_e = \frac{1}{n^2} \sum_{i=1}^{g} fi + f + i \tag{3}$$

where (fi+) is the total of row, and (f + i) is the total of column (*Viera & Garrett, 2005*).

When we use Cohen's Kappa (K), we can achieve one of the six types that represents in the Table 4.

For more details, the evaluation was done by a non-deaf coder who annotated each of the two ArSL video corpora on the basis of true and false independently. First, we measured the agreement between signer 1 and signer 2. Next, we measured the agreement between signer 1 and signer 3. Finally, we measured the agreement between signer 2 and signer 3. Therefore, by applying Cohen's Kappa statistical method to measure the agreement between each of the two videos in the ArSL *corpus*, we found the result represented in Table 5.

**Table 5 The result of evaluation using Cohen's Kappa.**

| | Number of observed agreements (%) | Number of agreements expected by chance (%) | Kappa (%) | Agreement |
|---|---|---|---|---|
| **Signer 1 & Signer 2** | 78.14 | 63.97 | 39 | **Fair Agreement** |
| **Signer 1 & Signer 3** | 78.14 | 66.39 | 35 | **Fair Agreement** |
| **Signer 2 & Signer 3** | *87.91* | *69.03* | *61* | **Good Agreement** |

Note:
The ArSL video corpus that was made by signer 2 or signer 3 got the best agreement based on Cohen's Kappa method for evaluation.

As we can see in (Table 5) The agreement between signer 2 and signer 3 reached about 60%, which is better than the others. The reason is that they are from the same school, the Deaf Association. However, signatory 1 is a volunteer who is not from their school. It should be noted that the first two are completely deaf, while the second is hard of hearing. In the validation module, in the future work, we will implement the ML technique using python as a programming language. To do this, we will use the ArSL video *corpus* that was made by signer 2 or signer 3. Which one got the best agreement based on Cohen's Kappa method for evaluation. Next, we will divide our data into training and testing datasets to measure accuracy and error rate.

## CONCLUSIONS AND FUTURE WORK

Through this research, we have reviewed the previous studies that have been conducted in the field of ArSL *corpus* generation as a standard sign language used by different countries. We have also illustrated the different areas in which researchers are conducting their efforts to create a data dictionary and annotated ArSL *corpus*, especially in Saudi Arabia. We have clarified the difficulties encountered in translating ArSL from a grammatical, semantic and syntactic perspective. How they affect the accuracy of translation and recognition. Finally, we described the ArSL dictionary for deaf drivers and explained the data collection processes to construct the videos in our *corpus*. These were recorded using a single camera and then verified using two methods. First, using the evaluation of four participants, experts in ArSL, two of whom were deaf. Second, by using Cohen's Kappa statistical method to measure the agreement between each of the two videos in the ArSL *corpus* recorded by three signers.

The created ArSL *corpus* provides opportunities to test various feature extraction methods and recognition techniques. Extending and validating the dataset using machine learning (ML) will be the focus of future work. In addition, this *corpus* will be used to design our proposed system to facilitate communication with deaf drivers.

### Funding

The authors received no funding for this work.

### Competing Interests

The authors declare that they have no competing interests.

## Author Contributions

- Samah Abbas conceived and designed the experiments, performed the experiments, analyzed the data, performed the computation work, prepared figures and/or tables, authored or reviewed drafts of the paper, and approved the final draft.
- Hassanin Al-Barhamtoshy analyzed the data, performed the computation work, prepared figures and/or tables, authored or reviewed drafts of the paper, and approved the final draft.
- Fahad Alotaibi performed the experiments, analyzed the data, authored or reviewed drafts of the paper, and approved the final draft.

## Data Availability

The raw data (videos) are available at Github:

- https://github.com/sabbas123/ArSL-video-corpus-1.git
- https://github.com/sabbas123/ArSL-video-corpus-2.git
- https://github.com/sabbas123/ArSL-video-corpus-3.git
- https://github.com/sabbas123/ArSL-video-corpus-4.git
- https://github.com/sabbas123/ArSL-video-corpus-5.git
- https://github.com/sabbas123/ArSL-video-corpus-6.git
- https://github.com/sabbas123/ArSL-video-corpus-7.git
- https://github.com/sabbas123/ArSL-video-corpus-8.git
- https://github.com/sabbas123/3-signer-evaluation_corpus1-welcome-.git
- https://github.com/sabbas123/3-signer-evaluation_corpus2-Directions-.git
- https://github.com/sabbas123/3-signer-evaluation_corpus3-Places-.git
- https://github.com/sabbas123/3-signer-evaluation_corpus4-Traffic-and-Transportation-.git
- https://github.com/sabbas123/3-signer-evaluation_corpus5-Driver-.git
- https://github.com/sabbas123/3-signer-evaluation_corpus6-Passenger-.git
- https://github.com/sabbas123/-sabbas123-3-signer-evaluation_corpus7-General-Words-.git
- https://github.com/sabbas123/3-signer-evaluation_corpus8-Amount-.git.

## Supplemental Information

Supplemental information for this article can be found online at http://dx.doi.org/10.7717/peerj-cs.741#supplemental-information.

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
