# Peer review of "Towards an Arabic Sign Language (ArSL) *corpus* for deaf drivers"

_PeerJ Computer Science, doi:10.7717/peerj-cs.741_

## Round 0.1 · original submission · Major Revisions

Kindly revise the manuscript as per the suggestions given by the reviewers.

·

Basic reporting

• English needs to be improved significantly. I suggest a professional proofread, so that your writing can reach and interest a wider audience. Here are some examples:
Lines 53-55: “…there are some researchers…provided their effort”
Line 139: “These complicities are explained below”
Line 286: “Feature Work”
Table 2: “أو”.

• Considerate language should be used when dealing with inclusiveness issues. Avoid, for example, the word "normal" as opposed to "deaf" (lines 23, 38, 40, 49, 58, 59, 60, 63).

• It is better to use acronyms after explaining their meaning. For example:
Lines 42-43: (ASL) American Sign Language, (BSL) British Sign Language, (ASL) Australian Sign Language, (ISL) Indian Sign Language and (ArSL) Arabic Sign Language
... could be changed to:
American Sign Language (ASL), British Sign Language (BSL), Australian Sign Language (Auslan), Indian Sign Language (ISL), and Arabic Sign Language (ArSL).
Note that the correct acronym for Australian Sign Language is Auslan.

• Avoid paragraphs that are too long, as it makes reading cumbersome (example: lines 81-118).

• The introduction and the background show context. However, lines 92-94 state that "For instance, Support Vector Machine (SVM) with a camera that gained 98.8 93% (Luqman & Mahmoud, 2017) (PCA) Principal Component Analysis, and (HMM) Hidden 94 Markov Model that achieved 99.9% accuracy (Ahmed & Aly, 2014)". Should these percentages be considered baselines? If so, this should be set from the abstract. In fact, the title should be reconsidered, as it indicates that it is an "Implementation", but then it becomes clear that it is a "Proposal". It should be clear from the outset whether the proposal is by any means compared against a baseline. It should be clear whether the proposal is by any means compared against a baseline. If the authors consider that this is not relevant, given that they focus on a new domain (public transport drivers and passengers), they should make it explicit. In the same way, whether it is an implementation or a proposal, it is good to say what it is. Eg: "implementation of a prototype of machine translation" or "proposal of a corpus with linguistic support".

• Although it is not a review article, it is good practice to briefly explain how the results presented in the Background have been produced. These search strings can enrich the Background section and it is left to the discretion of the authors to use them:
https://scholar.google.com/scholar?hl=es&as_sdt=1%2C5&as_ylo=2017&as_vis=1&q=intitle%3A%22sign+language%22+intitle%3A%22systematic%22&btnG=
https://scholar.google.com/scholar?hl=es&as_sdt=1%2C5&as_ylo=2017&as_vis=1&q=intitle%3A%22sign+languages%22+intitle%3A%22systematic%22&btnG=
https://scholar.google.com/scholar?hl=es&as_sdt=1%2C5&as_ylo=2017&as_vis=1&q=intitle%3A%22sign+language%22+intitle%3A%22state-of-the-art%22&btnG=

• The authorship of the figures must be clear. In some places, the author Abba is quoted and in others Abbas is quoted, causing confusion as to whether this paper is a continuation of earlier research. If so, you should make this clarification from the beginning and explain what is new with respect to the previous work. This is part of the context and it is recommended to place it at the beginning of the abstract.

Experimental design

• Methods are described with sufficient detail, in order to replicate.

• The journal requests in its guidelines that the research questions be well defined, relevant and meaningful. Please state in the Introduction what the research question is, which could be something like "How can a corpus be built by computer means for ArSL that is valid from a linguistic perspective?". The answer to this question must be made explicit in the Conclusions.

Validity of the findings

• The research findings appear to be valid. In any case, it is recommended to clearly indicate in line 279 why a WER of 10.23% is considered good (refer to the related literature).

• Underlying data have been provided. However, in "Chapter 8" concerning Amount, it is not clear why the word "Riyals" is omitted from number 5 onwards.

Additional comments

• The authors' research is extremely valuable since it addresses a problem of social inclusion in an adequate paradigm, which is that of providing tools for work and incorporation into the productive scheme. Innovation is evident within the public transport domain in Saudi Arabia. Changes are required to achieve the high publishing standards of PeerJ Computer Science, which is why this reviewer has attempted to do a thorough and explicit review.

·

Basic reporting

The introduction is clear and easy to follow, however, there are a few shortcomings.

Line 57, “deaf people have some difficulties in communicating with others while they are driving a vehicle like a car and the deaf or normal person sitting as a passenger”. This statement is confusing if this work aims at deaf people driving or deaf people sitting next to a driver? Authors can rephrase this for better understanding to the reader.

Literature survey on the ArSL recognition and machine translation is fine, but the novelty of this work is in the corpus.

A discussion on existing ArSL corpora like An Arabic Sign Language Corpus for Instructional Language in School (https://eprints.soton.ac.uk/271106/) and their limitations in the deaf driver domain can make the reader understand why this corpus is useful

Also, it would be good if authors can provide information related to deaf driver corpus in another sign language if there are any.

Fig. 6 didn’t have information on the y-axis. Assuming this as Word Error Rate (WER). Adding axis labels in the charts can ease the readers.

Experimental design

What are the problems with current approaches? Why deaf drivers can’t use the existing ArSL for communication during driving?

It has been mentioned that some of the words are derived from Saudi sign language and the rest of them are contextual to the domain. How are those words considered contextual to the domain? It has been mentioned that “Some of them are collected from the contextual domain of the normal conversation that done between taxi driver and passengers.”. This process needs to be more elaborate. The corpus should contain all the relevant words in the deaf driver domain without leaving any important information.

The corpus has 50% WER in the “Welcome” category but what’s the share of this “welcome” category in the whole corpus. Information about class distribution is important and needs to be included.

Validity of the findings

Good to see that corpus files made available for the community. However, annotations can’t be found. Any reason for this?

Additional comments

Line 282-285, please check the font style and make it uniform.

Add a citation or a URL as a footnote for the VEGAS video editor

There are a few grammatical errors in the English language used -For instance: In line 254, “For future work supporting, we added in…”.

---

## Round 0.2 · accepted · Accept

Kindly improve the English language presentation.

·

Basic reporting

I am satisfied with the important changes that the authors have made, in response to my recommendations.

Experimental design

I am satisfied with the important changes that the authors have made, in response to my recommendations.

Validity of the findings

I am satisfied with the important changes that the authors have made, in response to my recommendations.

Additional comments

I am satisfied with the important changes that the authors have made, in response to my recommendations. Clarifications of a cultural nature have also been valuable in order to give the paper a final go-ahead.

·

Basic reporting

Good to see the improvements in Introduction and literature review. The article is now in very good shape and easy to follow.

Experimental design

Methods are sufficient to replicate the results. No further changes are required.

Validity of the findings

No comment.

Additional comments

Interesting research and very useful to the community.

Reviewer 3 ·

Basic reporting

The manuscript need to be checked again with respect to issues on spelling and style. (for ex. Line No 32 - spelling of publish need to be checked again)

Uniformity in listing the references is expected.

Also it is suggested to cite the latest references in the manuscript.

In general, Basic reporting is ok.

Experimental design

This research focus on communicating with persons with hearing and speech impairment especially with drivers. The research question could have been presented well.

Methodology could have been presented in detail.

Validity of the findings

The time complexity of the proposed methodology could have been discussed.

Accuracy is not discussed.

Conclusion is ok.

Additional comments

This research focus on communicating with persons with hearing and speech impairment. what motivates the author to restrict with only drivers. Even the passengers with hearing and speech impairment need to communicate with a driver who is not like that. Title of the manuscript can be checked again with respect to this.

Line No 152 need to be checked. deaf people can communicate, but they cannot understand what others say.

The time complexity of the proposed methodology could have been discussed.

Accuracy could have been discussed.

The manuscript need to be checked again with respect to issues on spelling and style. (for ex. Line No 32 - spelling of publish need to be checked again)

Uniformity in listing the references and uniformity in citing the references is expected. Some of the articles cited are not listed in the reference. Also few papers in reference are not cited.

Also it is suggested to cite the latest references in the manuscript.